# Clinical and Rehabilitative Predictors of Peak Oxygen Uptake Following Cardiac Transplantation

**DOI:** 10.3390/jcm8010119

**Published:** 2019-01-19

**Authors:** Katelyn E. Uithoven, Joshua R. Smith, Jose R. Medina-Inojosa, Ray W. Squires, Erik H. Van Iterson, Thomas P. Olson

**Affiliations:** 1School of Kinesiology, University of Minnesota, Minneapolis, MN 55455, USA; 2Division of Preventive Cardiology, Department of Cardiovascular Medicine, Mayo Clinic, Rochester, MN 55905, USA; smith.joshua1@mayo.edu (J.R.S.); medinainojosa.jose@mayo.edu (J.R.M.-I.); squires.ray@mayo.edu (R.W.S.); olson.thomas2@mayo.edu (T.P.O.); 3Section of Preventive Cardiology and Rehabilitation, Cleveland Clinic, Cleveland, OH 44195, USA; vanitee@ccf.org

**Keywords:** peak oxygen uptake, cardiopulmonary exercise testing, cardiac rehabilitation, exercise capacity, postoperative care

## Abstract

The measurement of peak oxygen uptake (VO_2peak_) is an important metric for evaluating cardiac transplantation (HTx) eligibility. However, it is unclear which factors (e.g., recipient demographics, clinical parameters, cardiac rehabilitation (CR) participation) influence VO_2peak_ following HTx. Consecutive HTx patients with cardiopulmonary exercise testing (CPET) between 2007–2016 were included. VO_2peak_ was measured from CPET standard protocol. Regression analyses determined predictors of the highest post-HTx VO_2peak_ (i.e., quartile 4: VO_2peak_ > 20.1 mL/kg/min). One hundred-forty HTx patients (women: *n* = 41 (29%), age: 52 ± 12 years, body mass index (BMI): 27 ± 5 kg/m^2^) were included. History of diabetes (Odds Ratio (OR): 0.17, 95% Confidence Interval (CI): 0.04–0.77, *p* = 0.021), history of dyslipidemia (OR: 0.42, 95% CI: 0.19–0.93, *p* = 0.032), BMI (OR: 0.90, 95% CI: 0.82–0.99, *p* = 0.022), hemoglobin (OR: 1.29, 95% CI: 1.04–1.61, *p* = 0.020), white blood cell count (OR: 0.81, 95% CI: 0.66–0.98, *p* = 0.033), CR exercise sessions (OR: 1.10, 95% CI: 1.04–1.15, *p* < 0.001), and pre-HTx VO_2peak_ (OR: 1.17, 95% CI: 1.07–1.29, *p* = 0.001) were significant predictors. Multivariate analysis showed CR exercise sessions (OR: 1.10, 95% CI: 1.03–1.16, *p* = 0.002), and pre-HTx VO_2peak_ (OR: 1.16, 95% CI: 1.04–1.30, *p* = 0.007) were independently predictive of higher post-HTx VO_2peak_. Pre-HTx VO_2peak_ and CR exercise sessions are predictive of a greater VO_2peak_ following HTx. These data highlight the importance of CR exercise session attendance and pre-HTx fitness in predicting VO_2peak_ post-HTx.

## 1. Introduction

Peak oxygen uptake (VO_2peak_) is an important metric for cardiac transplantation (HTx) eligibility. The measurement of exercise capacity using VO_2peak_ obtained from cardiopulmonary exercise testing (CPET) is one criteria used for determination of transplant eligibility; with a value of ≤14 mL/kg/min considered transplant eligible [1,2,3,4,5]. The association between higher VO_2peak_ and reduced hospitalizations and mortality risk has been shown in both the heart failure (HF) and HTx population, with even modest improvements in VO_2peak_ associated with improved outcomes [6,7,8,9]. Because of this, investigating demographic and clinical factors that are predictive of post-HTx VO_2peak_ has significant implications not only for functional capacity but also long-term survival.

Previous studies have investigated demographic and clinical determinants of VO_2peak_ in various populations [10,11,12,13,14,15,16,17,18]. For example, the predictive influence of age, sex, and body mass index (BMI) on VO_2peak_ has been shown in healthy populations [16,17]. In HTx patients, these variables have also been shown to be prominent predictors of post-HTx VO_2peak_ along with chronotropic reserve, donor age, and time from HTx [11,13,14,18]. However, a limitation of these previous studies is the small sample sizes used (*n* = 60–95) [11,13,14,18]. In addition, recent work has indicated that participation in cardiac rehabilitation (CR) in HTx postoperative care has been shown to be related to improvements in VO_2peak_ [15,19,20]; however, it is unclear if CR participation is predictive of a greater VO_2peak_ following HTx.

Therefore, the purpose of this study was to investigate whether pre-HTx clinical characteristics and/or postoperative CR exercise session attendance provide utility in predicting VO_2peak_ following HTx. Based on previous studies on the relationship between CR involvement and VO_2peak_ [15,19,20], we hypothesize that CR will surpass other predictive factors of post-HTx VO_2peak_ in HTx patients.

## 2. Experimental Section

### 2.1. Participants and Study Design

A retrospective, single-center study cohort design evaluated consecutive adult HTx patients who performed symptom-limited CPET prior to HTx (pre-HTx) and following HTx (post-HTx) between the years of 2007–2016. Demographic and clinical characteristics were obtained from an institutional database. Inclusion criteria included completion of pre-HTx CPET within 24 months prior to procedural date and post-HTX CPET within 1-year of HTx. Patients were excluded if they lacked CR exercise session data or had incomplete CPET data. Of the 204 HTx patients, 140 were analyzed in this study (Figure 1). This study was approved by the Mayo Clinic Institutional Review Board (IRB #15-007965) and followed research authorization protocol for the use of medical records as required by the state of Minnesota [21].

### 2.2. Clinical Characteristics

Clinical baseline information from the time of HTx procedural date was obtained via medical record extraction. Demographic data along with previous disease history, previous left ventricular assist device (LVAD), current laboratory measurements (i.e., hemoglobin, hematocrit, white blood cell count, and creatinine), indication for HTx (i.e., restrictive cardiomyopathy, dilated cardiomyopathy, hypertrophic cardiomyopathy, ischemic cardiomyopathy, or other), and pre-HTx medication status for the following: angiotensin-converting enzyme (ACE) inhibitor, amiodarone, aspirin, beta blocker, calcium channel blocker, and diuretic were extracted from the procedural sedation assessment at the time of HTx. For the purpose of monitoring data correctness, two investigators independently reviewed a random sampling of medical record charts.

### 2.3. Cardiac Rehabilitation Participation

Patients included in this study were referred for CR participation and attended at least one documented session following HTx. Medical records were examined to determine CR attendance specifically relating to postoperative HTx care versus CR for any cardiac-related event. As the initial visit for CR typically involves orientation procedures with little to no exercise involvement, this was not assessed in this study. Only those CR sessions with documented exercise participation were included for analysis. All exercise sessions were supervised throughout activity by clinical exercise physiologists with cardiologist oversight. During the course of CR participation patients performed 20–45 min of aerobic activity in a monitored setting, with the usual addition of strength training components for 10–15 min. In addition to CR attendance patients were encouraged to partake in light to moderate physical activity for at least 30 min on all days of the week. Guidance regarding nutrition, medication management, stress management, and depressive symptom management were all components of the comprehensive approach of the CR program. Social support was also provided in this environment as patients all engaged in group education/learning activities throughout CR.

### 2.4. Cardiopulmonary Exercise Testing Procedures

Clinical exercise physiologists conducted the clinically-indicated CPET with cardiologist oversight. An institutionally designed protocol was performed on a motorized treadmill (GE Case, Milwaukee, WI, USA) with workload increasing every 2 min until volitional fatigue [22,23]. Heart rhythm and heart rate (HR) were continuously monitored via 12-lead electrocardiogram during exercise and recovery. Prior to the start of exercise and at the last 30 s of each stage systolic blood pressure (SBP) and diastolic blood pressure (DBP) were measured via manual sphygmomanometry. Flow and gas exchange variables were measured during exercise using indirect calorimetry (MGC Diagnostics, St. Paul, MN, USA). Peak values were obtained by averaging the last 30 s of the CPET data collection. Measured gas exchange variables included VO_2peak_, carbon dioxide production (VCO_2_), minute ventilation (V_E_), and respiratory exchange ratio (RER). The value of V_E_/VCO_2_ slope was determined from rest-peak values for V_E_ and VCO_2_ and %predicted VO_2peak_ [24] was calculated. Peak values were chosen for V_E_/VCO_2_ slope as they have been shown to have superior prognostic value as opposed to the pre-ventilatory threshold slope equation [25]. Patients were monitored closely for development of significant dysrhythmia, electrocardiographic abnormalities, and/or abnormal blood pressure responses (e.g., symptomatic hypotension). Cardiac medications were not withheld prior to CPET to maximize patient safety and generalizability of the data.

### 2.5. Statistical Analysis

Data are reported as mean ± standard deviation (SD) or frequency (percentage) where applicable. Statistical analysis was conducted using SPSS (version 22.0, Chicago, IL, USA) and JMP (JMP, Cary, NC, USA). The Student’s *t* test was used to compare pre-HTx CPET variables to post-HTx CPET variables. As previously described [26,27], univariate binary regression analysis was used to assess individual predictors for the highest post-HTx VO_2peak_ (i.e., quartile 4) based on VO_2peak_ data presented herein. Quartile 4 = VO_2peak_ > 20.1 mL/kg/min, quartile 3 = VO_2peak_ > 17.4 mL/kg/min, quartile 2 = VO_2peak_ > 14.3 mL/kg/min, and quartile 1 = VO_2peak_ ≤ 14.3 mL/kg/min (total range = 24.5 mL/kg/min). A multivariate regression analysis was then performed including all significant univariate variables. Three additional adjustment analyses were conducted to account for potential factors involved in exercise capacity. Model 1 was adjusted for demographic influencers of exercise capacity (i.e., age and sex), Model 2 was adjusted for demographic and clinical influencers of exercise capacity (i.e., age, sex, BMI, history of diabetes, and hemoglobin), and Model 3 included all significant univariate predictors for multivariate analysis adjusted for age and sex. The receiver operating characteristic (ROC) curve model was assessed to establish the area under the curve (AUC) for predicting quartile 4 (i.e., the highest VO_2peak_) and the cut-off for CR for maximal post-HTx VO_2peak_ benefits. Least squares linear regression analysis was performed between pre-HTx VO_2peak_ and post-HTx VO_2peak_, with reporting of the Pearson product-moment correlation coefficient (r) as an indicator of the strength of association between before and after measurements. For all analyses, statistical significance was set at an alpha level of *p* < 0.05.

## 3. Results

### 3.1. Patient Population and Clinical Characteristics

Demographic and clinical characteristics of the patients included are presented in Table 1. Of the 140 HTx patients analyzed in this study the mean age was 52 ± 12 years, mean BMI was 27 ± 5 kg/m^2^, and *n* = 41 (29%) were women. All data shown in Table 1 was obtained from each patient within 24 months prior to procedural data. The mean number of CR sessions attended was 18 ± 9, with a range of 2–36 sessions.

### 3.2. Exercise Testing Data Prior to and Following Transplant

Peak exercise testing data for CPET pre-HTx versus post-HTx are shown in Table 2. All variables included show significant improvements from pre-HTx to post-HTx, with V_E_/VCO_2_ slope decreasing while all other variables increased. Notably, relative VO_2peak_ increased substantially from pre- to post-HTx (12.9 ± 4.4 vs. 17.5 ± 4.6 mL/kg/min, *p* < 0.001). A significant positive correlation was present between pre-HTx VO_2peak_ and post-HTx VO_2peak_ (Figure 2) (*r* = 0.47, *p* < 0.01). Additionally, this correlation between pre and post-HTx VO_2peak_ was further analyzed with adjustment for the number of CR exercise sessions attended and remained significant (*p* < 0.001).

### 3.3. Predictors of VO_2peak_

Univariate regression analysis indicated that BMI, history of diabetes, history of dyslipidemia, hemoglobin, white blood cell count, CR exercise sessions, and pre-HTx VO_2peak_ were significant predictors of higher VO_2peak_ post-HTx (Table 3). The significance of pre-HTx VO_2peak_ and CR exercise sessions on post-HTx VO_2peak_ remained after adjustment for demographic influencers of exercise capacity (age and sex) as shown in Model 1, and adjustment for demographic and clinical influencers of exercise capacity (age, sex, BMI, history of diabetes, hemoglobin, peak HR, and HR recovery (HRR)) as shown in Model 2 (Table 4). 

All significant univariate predictors were included into the multivariate regression analysis with adjustment for age and sex. This analysis (as shown in Model 3) indicated that CR exercise sessions and pre-HTx VO_2peak_ were the independent predictors of higher post-HTx VO_2peak_ (Table 4). When CR exercise session attendance was divided into two groups by the median attendance of 18 sessions, those who attended ≥18 sessions had significantly higher post-HTx VO_2peak_ compared to those with <18 sessions attended (≥18 sessions: 18.8 ± 4.8 vs. <18 sessions: 16.4 ± 4.3, *p* < 0.01) (Figure 3). The value of 18 sessions was also determined to be an appropriate cutoff for CR attendance necessary to maximize benefits in post-HTx VO_2peak_ (AUC: 0.692, specificity: 0.549, sensitivity: 0.684, *p* < 0.001).

## 4. Discussion

The purpose of this study was to determine whether pre-HTx clinical characteristics and/or postoperative participation in CR provide utility in predicting VO_2peak_ following HTx. Pre-HTx VO_2peak_ and the number of CR exercise sessions attended were the only variables predictive of greater VO_2peak_ following HTx. These metrics independently predicted higher post-HTx VO_2peak_ even after adjusting for other factors of exercise capacity, surpassing other significant univariate factors. Our data provide new evidence of the critical benefits of CR in improving functional capacity following HTx.

Previous work has evaluated potential predictors of VO_2peak_ in various populations [10,11,12,13,14,17,18,28,29]. In healthy subjects the general characteristics of younger age, male sex, and training state have been shown to be associated with improved exercise capacity [17]. For those with cardiovascular disease (CVD), several studies measuring exercise capacity directly by VO_2peak_ or indirectly using predicted METS have shown overall that age and baseline fitness are the strongest predictors of VO_2peak_ [10,28,29]. The predictive quality of age also remains in the HTx population [11,13,14]. One of the first studies to evaluate predictors of VO_2peak_ specifically for HTx patients found that chronotropic reserve, donor age, recipient age, and time from HTx were the most significant predictors of VO_2peak_ post-HTx [11]. This study, however, collected CPET data in the broad range of 1–100 months following HTx, which limits the specificity of temporal cardiopulmonary adaptation post-HTx. Although not significant in multivariate analysis, clinical history of diabetes and dyslipidemia were significant univariate predictors in this study, emphasizing the importance of multimorbidity management [30,31,32,33]. More recent studies in HTx patients have shown that the demographic components of sex, BMI, and age are prominent predictors of VO_2peak_ [13,14,18]. Table 5 provides information on four specific studies particularly relevant to the findings of the current study examining predictors of VO_2peak_ in HTx patients.

Although the findings of these studies concur with our study regarding the predictive quality of BMI, CR exercise session attendance and baseline fitness as assessed by pre-HTx VO_2peak_ were the only independent predictors of post-HTx VO_2peak_. Baseline fitness levels have previously been shown to predict VO_2peak_ in CVD patients [28,29], yet this study is the first to show this relationship in HTx patients. A significant correlation was present between pre-HTx VO_2peak_ and post-HTx VO_2peak_ in our study, as those with the highest pre-HTx VO_2peak_ were more likely to have higher VO_2peak_ measurements following HTx. These results highlight the importance of maintaining or reaching a higher functional capacity leading up to HTx, as baseline values predict postoperative VO_2peak_ levels.

The predictive impact of CR involvement following HTx on achieving higher VO_2peak_ values was a unique component to this study. Our findings clearly demonstrate the importance of postoperative CR participation in improving functional capacity and implications of survival following HTx. Specifically following HTx, the benefits of CR involvement are widespread for this clinically unique population; including counteracting marked deconditioning and skeletal muscle weakness associated with end-stage HF, corticosteroid treatment, and surgical recovery [18,34,35,36]. Although the predictive quality of CR session attendance has not been previously reported, the increase in VO_2peak_ following CR in HTx patients is well-documented [15,19,20,37,38]. The importance of CR involvement on VO_2peak_ following HTx is critical, as it elicits significantly greater increases in VO_2peak_ compared to at-home therapy [37]. One such study reported an increase of 3.6 mL/kg/min in VO_2peak_ following a 12-week CR program after HTx [19]. Additionally, a recent review evaluating CR exercise in HTx patients that encompassed 10 randomized controlled trials with a total of 300 patients found that exercise capacity was, on average, 2.49 mL/kg/min higher for those who exercised [15]. Our study showed a mean increase of 4.6 mL/kg/min (equivalent to over 1.5 METS) from pre- to post-HTx, which has substantial meaning with regard to improved quality of life and survival. Further, those who attended >18 sessions demonstrated a post-HTx VO_2peak_ of 2.4 mL/kg/min higher than those who attended <18 sessions (i.e., 15% greater). Studies in HF have found that an increased VO_2peak_ is associated with better outcomes (i.e., primary endpoints of all-cause hospitalizations and/or mortality) [6,7,39,40]. One such study found a 5% lower risk of all-cause mortality or hospitalization for every 6% increase in VO_2peak_, thereby highlighting the importance of even seemingly small improvements in VO_2peak_ on long-term clinical outcomes [7]. It should be noted, however, that despite significant functional improvements observed following HTx, VO_2peak_ values still remain abnormal in HTx patients compared to age-matched control subjects [34,41,42]. Further understanding of the demographic, clinical, and rehabilitative components that predict post-HTx VO_2peak_ values offers greater insight into the complex cardiopulmonary and peripheral adaptations occurring following HTx.

It is important to recognize the retrospective observational nature of the present study design. Additional research in this area is encouraged to confirm these results in a prospective fashion. The distribution of post-HTx medications could potentially influence CPET measurements, therefore this should be taken into account as a limitation to the application of these results. Further, data pertaining to donor information (e.g., donor age, donor sex) and surgical procedural notes (e.g., cold ischemic time) were unavailable for this study and may be important additional metrics to include in follow-up studies.

## 5. Conclusions

In summary, HTx patients demonstrated substantial improvements in VO_2peak_ following HTx. The only factors that provided significant predictive value of higher post-HTx VO_2peak_ values were CR exercise session attendance and pre-HTx VO_2peak_, even after adjustment for other univariate predictors of post-HTx exercise capacity. These data demonstrate the influential role of sustaining and/or improving cardiorespiratory fitness leading up to and following HTx, specifically in the CR setting.

## Figures and Tables

**Figure 1 jcm-08-00119-f001:**
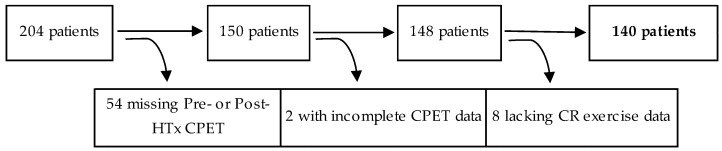
Flowchart for patient inclusion and exclusion. Of the initially identified 204 HTx patients, 54 patients lacked a pre-HTx or post-HTx CPET, 2 patients had incomplete CPET data, and 8 patients were lacking CR exercise session data, resulting in 140 patients for study analysis. HTx, cardiac transplantation; CPET, cardiopulmonary exercise testing.

**Figure 2 jcm-08-00119-f002:**
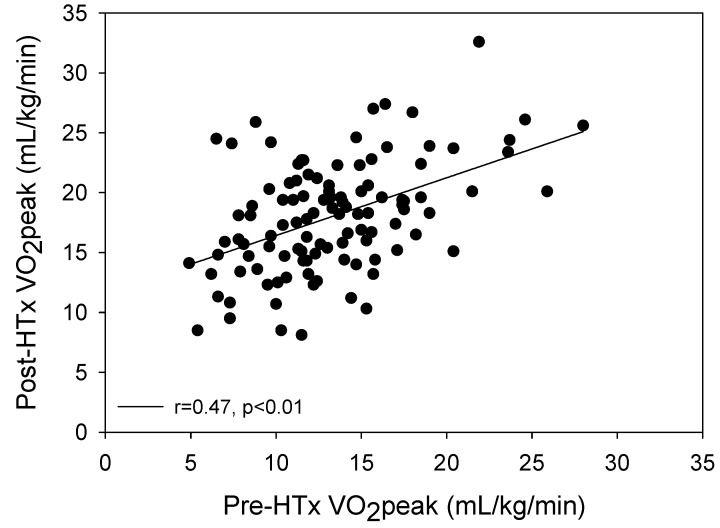
Relationship between measurements of relative VO_2peak_ from pre-HTx to post-HTx. A significant positive correlation was present between pre-HTx VO_2peak_ and post-HTx VO_2peak_ (*r* = 0.47, *p* < 0.01). Post-HTx: following cardiac transplantation; Pre-HTx: prior to cardiac transplantation; VO_2peak_: peak oxygen uptake.

**Figure 3 jcm-08-00119-f003:**
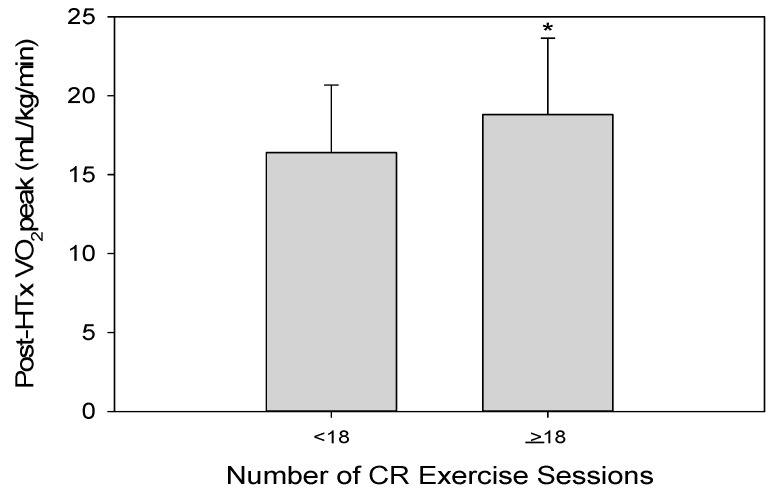
Post-HTx VO_2peak_ based on median CR exercise session attendance. Those HTx patients who attended ≥18 CR exercise sessions had significantly higher post-HTx VO_2peak_ compared to those with <18 CR exercise sessions attended (≥18 sessions: 18.8 ± 4.8 vs. <18 sessions: 16.4 ± 4.3, *p* < 0.01) *, significantly higher than <18 CR sessions. CR: cardiac rehabilitation; Post-HTx: following cardiac transplantation; VO_2peak_: peak oxygen uptake.

**Table 1 jcm-08-00119-t001:** Recipient demographics and clinical characteristics.

*n*	140
Age (years)	52 ± 12
Sex (Female)	41 (29)
Height (cm)	172 ± 14
Weight (kg)	83 ± 19
BSA (m^2^)	2.0 ± 0.2
BMI (kg/m^2^)	27.1 ± 4.6
History of Diabetes	31 (22.1)
History of Smoking	52 (37.1)
History of Dyslipidemia	74 (52.9)
History of Hypertension	63 (45)
Previous LVAD	27 (19.3)
Indication for Heart Transplant	
Restrictive Cardiomyopathy	29 (20.7)
Dilated Cardiomyopathy	53 (37.9)
Hypertrophic Cardiomyopathy	8 (5.6)
Ischemic Cardiomyopathy	25 (17.9)
Other	25 (17.9)
Labs	
Hemoglobin (g/dL)	12.0 ± 2.0
Hematocrit (%)	36 ± 6
White Blood Cell Count (10^9^/L)	7.5 ± 3.0
Creatinine (mg/dL)	1.4 ± 1.0
Medications *	
ACE Inhibitor	52 (37.1)
Amiodarone	12 (8.6)
Aspirin	65 (46.4)
Beta Blocker	103 (73.6)
Calcium Channel Blocker	3 (2.1)
Diuretic	92 (65.7)
Number of CR Exercise Sessions	18 ± 9

Note: BMI, body mass index; BSA, body surface area; CR, cardiac rehabilitation; LVAD, left ventricular assist device. All data are presented as mean ± standard deviation or frequency (percentage). *, medication distributions are prior to cardiac transplantation procedure; ACE, angiotensin converting enzyme.

**Table 2 jcm-08-00119-t002:** Peak exercise testing values prior to and following cardiac transplantation.

	Pre-HTx	Post-HTx	*p*-Value
*n*	140	140	
Exercise time (min)	5.3 ± 1.7	6.7 ± 1.7	<0.001
METS	4.9 ± 1.8	6.4 ± 1.8	<0.001
Absolute VO_2peak_ (L/min)	1.1 ± 0.4	1.4 ± 0.4	<0.001
VO_2peak_ % Predicted (%)	42 ± 14	58 ± 17	<0.001
Relative VO_2peak_ (mL/kg/min)	12.9 ± 4.4	17.5 ± 4.7	<0.001
V_E_/VCO_2_ slope	42 ± 12	37 ± 6	<0.001
VCO_2_ (L/min)	1.2 ± 0.5	1.7 ± 0.5	<0.001
RER	1.15 ± 0.13	1.21 ± 0.12	<0.001
SBP (mmHg)	105 ± 25	145 ± 30	<0.001
DBP (mmHg)	60 ± 10	67 ± 11	<0.001
HR (bpm)	110 ± 21	125 ± 20	<0.001

METS, metabolic equivalents; RER, respiratory exchange ratio; SBP, systolic blood pressure; DBP, diastolic blood pressure; HR, heart rate; VCO_2_, production of carbon dioxide; V_E_, minute ventilation; V_E_/VCO_2_ slope, ventilatory efficiency; VO_2peak_, peak oxygen uptake.

**Table 3 jcm-08-00119-t003:** Univariate regression analysis for predictors of VO_2peak_.

Variable	Univariate Analysis		
	OR	95% CI	*p*-Value
Age (years)	0.978	0.949–1.009	0.170
Sex (female)	0.457	0.173–1.209	0.115
BMI (kg/m^2^)	0.896	0.815–0.985	0.022
History of Diabetes	0.174	0.039–0.772	0.021
History of Smoking	0.805	0.354–1.831	0.605
History of Hypertension	0.741	0.335–1.641	0.460
History of Dyslipidemia	0.415	0.185–0.929	0.032
History of LVAD	1.171	0.446–3.076	0.748
Beta Blocker Medication	0.642	0.275–1.499	0.306
Indication for HTx			
Restrictive Cardiomyopathy	0.812	0.299–2.201	0.682
Dilated Cardiomyopathy	0.920	0.409–2.066	0.840
Hypertrophic Cardiomyopathy	2.040	0.461–9.035	0.348
Ischemic Cardiomyopathy	0.565	0.179–1.782	0.330
Other	1.694	0.655–4.380	0.277
Labs			
Hemoglobin (g/dL)	1.294	1.041–1.608	0.020
Hematocrit (%)	1.063	0.985–1.146	0.115
White Blood Cell Count (10^9^/L)	0.806	0.662–0.982	0.033
Creatinine (mg/dL)	0.402	0.155–1.039	0.060
Pre-HTx CPET Data			
Peak SBP (mmHg)	0.998	0.971–1.005	0.158
Heart Rate Recovery (bpm)	1.021	0.987–1.056	0.225
Relative VO_2peak_ (mL/kg/min)	1.174	1.070–1.289	0.001
CR Exercise Sessions	1.095	1.041–1.152	<0.001

OR, odds ratio; CI, confidence interval; BMI, body mass index; CR, cardiac rehabilitation; HTx, cardiac transplantation; LVAD, left ventricular assist device; SBP, systolic blood pressure, VO_2peak_, peak oxygen uptake.

**Table 4 jcm-08-00119-t004:** Adjusted and multivariate regression analysis for predictors of Post-HTx VO_2peak_.

Variable	Model 1			Model 2			Model 3		
	OR	95% CI	*p*-Value	OR	95% CI	*p*-Value	OR	95% CI	*p*-Value
Age (years)	0.968	0.932–1.006	0.094	0.970	0.933–1.009	0.130	0.962	0.922–1.004	0.077
Sex (female)	0.395	0.128–1.217	0.106	0.284	0.080–1.010	0.052	0.292	0.087–0.986	0.047
BMI (kg/m^2^)				0.889	0.791–0.999	0.049	0.097	0.811–1.018	0.097
History of Diabetes				3.050	0.602–15.450	0.178	2.760	0.635–11.995	0.176
History of Dyslipidemia							0.880	0.315–2.465	0.807
Labs									
Hemoglobin (g/dL)				1.172	0.930–1.478	0.178	1.045	0.829–1.31	0.710
White Blood Cell Count (10^9^/L)							0.796	0.620–1.021	0.072
Pre-HTx CPET Data									
Relative VO_2peak_ (mL/kg/min)	1.145	1.037–1.264	0.007	1.110	1.002–1.231	0.047	1.206	1.068–1.361	0.002
Peak HR (bpm)				0.979	0.951–1.007	0.147			
HRR (bpm)				1.008	0.959–1.059	0.764			
CR Exercise Sessions	1.102	1.037–1.264	<0.001	1.095	1.037–1.157	0.001	1.103	1.042–1.167	0.001

OR, odds ratio; CI, confidence interval; BMI, body mass index; CPET, cardiopulmonary exercise testing; CR, cardiac rehabilitation; HR, heart rate; HRR, heart rate recovery; LVAD, left ventricular assist device; SBP, systolic blood pressure; VO_2peak_, peak oxygen uptake. Model 1: Adjustment for age and sex. Model 2: Adjustment for age, sex, and influencers of exercise capacity (body mass index, history of diabetes, hemoglobin, peak HR, and HRR). Model 3: Significant univariate predictors in multivariate analysis with adjustment for age and sex.

**Table 5 jcm-08-00119-t005:** Predictors of VO_2peak_ in heart transplant patients.

Study Group	*n*	Age (yrs)	Predictors of VO_2peak_	Time from Transplant	Post-Transplant VO_2peak_ (mL/kg/min)
Douard et al. 1997 [11]	85	52 ± 12	Chronotropic reserve, time from transplantation, age of donor, age of patient	1–100 months	21.1 ± 6.0
Leung et al. 2003 [13]	95	48 ± 14	Age, sex, height, and weight (alternatively, body mass index)	12 months	19.9 ± 4.8
Nytrøen et al. 2012 [18]	51	52 ± 16	Muscular exercise capacity and body fat	1–8 years	Group 1: 23.1 ± 3.7; Group 2: 32.6 ± 4.4
Carvalho et al. 2015 [14]	60	48 ± 15	Age, sex, body mass index, heart rate reserve, and left atrium diameter	64 ± 54 months	unspecified

Note: VO_2peak_, peak oxygen uptake; yrs, years. All data are presented as mean ± standard deviation unless otherwise specified.

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
