# Peer review of "Clinical and Rehabilitative Predictors of Peak Oxygen Uptake Following Cardiac Transplantation"

_jcm, 2019, doi:10.3390/jcm8010119_

Reviewer 1 Report

GENERAL COMMENTS

This retrospective single-center study aimed to examine independent predictors of peak oxygen uptake (VO2peak) after cardiac transplantation. The cohort consisted of 140 patients that underwent cardiac transplantation (mostly men) during the period of 2007-2016. Study authors found that diabetes mellitus, history of dyslipidemia, body mass index, hemoglobin count, WBC, cardiac rehabilitation exercise sessions and pre-transplant VO2peak were independently associated with VO2peak after transplantation, however, in univariate binary regression analysis. Multivariable analysis showed that CR exercise sessions and pre-transplantation VO2peak were independent predictors of the higher post-transplantation VO2peak implicating that CR exercise session attendance is an important part of better post-transplantation outcomes, along with pre-transplantation fitness, thus being able to positively impact on post-transplantation VO2peak. I would wish to congratulate authors for conducting this important study, in this particularly challenging patient group.

 Ethical/legal disclosures:

Legal and ethics disclosures are present and described appropriately.

Statistical methods:

Statistical methods are sufficiently described and generally appropriate but could use some improvements.

 SPECIFIC COMMENTS

Intro section/Abstract:

Page 1. Define HTx as a „cardiac transplantation“ and VO2peak in the Abstract. Please make sure that you consistently explain abbreviations at the first mention in the text.

Introduction sufficiently describes study goals and the current state of knowledge in the respective arena.

METHODS

Page 2. Experimental Section – please break down exactly why 64 out of 204 HTx patients were not included in the study and did these patients significantly differ in contrast to those that were included in the study. This is important in order to avoid selection bias. A flowchart depicting inclusion and exclusion criteria with a specific breakdown of reasons for ineligibility would be recommended for addition to the manuscript.

Page 3. Cardiac rehabilitation participation – authors did a good thing by defining what CR participation actually means and included only patients that completed at least one session that included documented exercise.

Page 3. Statistical analysis. Please describe clearly how quartiles were derived. Was it from your own population and your own VO2 peak values, or you used quartile division cut-offs based on some previous research? Just make this clear and consistent.

RESULTS

Page 7. Please re-run statistical model number 3, as presented in Table 4. and also include sex and age in the model and report odds ratios for peak relative VO2peak (pre-HTx CPET data) and CR exercise sessions. These types of analyses should indeed be, at least, sex and age-adjusted, regardless of sex and age being insignificant in the univariate regression model.

Authors confirmed that pre-htX VO2peak correlated with post-HTx VO2peak, although I would also try to run this type of analysis as multiple linear regression and adjust it for a number of attended CR sessions. You can run this analysis and present this data as a supplement, perhaps.

DISCUSSION

In addition to previously elaborated findings, authors also found that there were statistically significant differences in post-HTx VO2peak between patients with under 18 of CR sessions vs. those with >18 sessions. They quote the difference of 18.8±4.8 for >18 sessions vs. 16.4±4.3 for<18 sessions, p<0.01. While this is an undoubtedly better result, it would be worthwhile to discuss how these differences might truly impact on symptoms, patient well-being and quality of life and if there are works in the literature that comment on the magnitude of VO2peak changes and improvement in the quality of life. I think this would be a great addition to the discussion.

 Study shortcomings and weaknesses

One of the more significant shortcomings of this work is the lack of post-transplant medication data used prior to the second measurement of VO2peak.  It is logical that medications used post-translation could indeed significantly impact on measured outcomes and this should be further emphasized as a limitation in the study. It should be clearly stated in Table 1. what do you mean by Medications. Is this prior to transplantation or drugs used after the transplantation prior to 2nd measurement? If medication intake refers to post-transplantation medication intake then regression analyses should be adjusted for medications. Please elaborate on this to greater detail.

Another shortcoming is the missing data on donor information such as donor age and donor sex which could certainly impact clinical outcomes.

This is an important and well-conducted study on a fairly high number of cardiac transplant patients which is always a particularly vulnerable and somewhat exclusive population.

 I would advise authors to redo some of the statistical analyses and particularly assess the impact of a number of CR sessions (as a continuous or ordinal variable) on the VO2peak. It would be beneficial to know at what particular cut-off does CR yield maximum benefits. This cannot be ascertained if CR is incorporated into the regression model as a dichotomous variable. I would also advise inspecting the correlation between post-HTx VO2peak and number of CR sessions, similarly as reported for pre-HTx VO2 peak vs. post-HTx VO2 peak presented in Figure 1.

Author Response

Response to Reviewer 1 Comments

 We would like to thank this reviewer for providing thoughtful insight and thorough review of our manuscript. After amending this manuscript based on the reviewer’s comments, we feel that this manuscript is substantially improved. Below is a point-by-point response to each of the reviewer’s questions and suggestions:

 GENERAL COMMENTS

This retrospective single-center study aimed to examine independent predictors of peak oxygen uptake (VO2peak) after cardiac transplantation. The cohort consisted of 140 patients that underwent cardiac transplantation (mostly men) during the period of 2007-2016. Study authors found that diabetes mellitus, history of dyslipidemia, body mass index, hemoglobin count, WBC, cardiac rehabilitation exercise sessions and pre-transplant VO2peak were independently associated with VO2peak after transplantation, however, in univariate binary regression analysis. Multivariable analysis showed that CR exercise sessions and pre-transplantation VO2peak were independent predictors of the higher post-transplantation VO2peak implicating that CR exercise session attendance is an important part of better post-transplantation outcomes, along with pre-transplantation fitness, thus being able to positively impact on post-transplantation VO2peak. I would wish to congratulate authors for conducting this important study, in this particularly challenging patient group.

The authors would like to thank the reviewer for their favorable comments.

Ethical/legal disclosures:

Legal and ethics disclosures are present and described appropriately.

Statistical methods:

Statistical methods are sufficiently described and generally appropriate but could use some improvements.

 SPECIFIC COMMENTS

Intro section/Abstract:

Page 1. Define HTx as a “cardiac transplantation” and VO2peak in the Abstract. Please make sure that you consistently explain abbreviations at the first mention in the text.

We appreciate this comment and have ensured all abbreviations are described at first use.

Introduction sufficiently describes study goals and the current state of knowledge in the respective arena.

METHODS

Page 2. Experimental Section – please break down exactly why 64 out of 204 HTx patients were not included in the study and did these patients significantly differ in contrast to those that were included in the study. This is important in order to avoid selection bias. A flowchart depicting inclusion and exclusion criteria with a specific breakdown of reasons for ineligibility would be recommended for addition to the manuscript.

We agree that this is an important point to clarify. An additional description has now been added in the Methods section to provide a specific breakdown of the inclusion and exclusion of patients. Additionally, a flowchart has been added to the Methods section for additional clarity:

 Figure 1. Flowchart for Patient Inclusion and Exclusion. Of the initially identified 204 HTx patients, 54 patients lacked a pre-HTx or post-HTx CPET, 2 patients had incomplete CPET data, and 8 patients were lacking CR exercise session data, resulting in 140 patients for study analysis.

Page 3. Cardiac rehabilitation participation – authors did a good thing by defining what CR participation actually means and included only patients that completed at least one session that included documented exercise.

Thank you for this comment.

Page 3. Statistical analysis. Please describe clearly how quartiles were derived. Was it from your own population and your own VO2 peak values, or you used quartile division cut-offs based on some previous research? Just make this clear and consistent.

We have now added additional detail regarding quartile divisions to further clarify this important point:

“As previously described [25,26], univariate binary regression analysis was used to assess individual predictors for the highest post-HTx VO2peak (i.e. quartile 4) based on VO2peak data presented herein.”

 RESULTS

Page 7. Please re-run statistical model number 3, as presented in Table 4. and also include sex and age in the model and report odds ratios for peak relative VO2peak (pre-HTx CPET data) and CR exercise sessions. These types of analyses should indeed be, at least, sex and age-adjusted, regardless of sex and age being insignificant in the univariate regression model.

We appreciate this comment and have now re-analyzed Model 3 with the addition of age and sex. The Results section as well as Table 4 have been adjusted accordingly.

Authors confirmed that pre-HTx VO2peak correlated with post-HTx VO2peak, although I would also try to run this type of analysis as multiple linear regression and adjust it for a number of attended CR sessions. You can run this analysis and present this data as a supplement, perhaps.

Thank you for this suggestion. The correlation between pre-HTx VO2peak and post-HTx VO2peak was analyzed with adjustment for CR exercise session attendance and remained significant (p<0.001). This finding was added to the Results in Section 3.2:

“Additionally, this correlation between pre and post-HTx VO2peak was further analyzed with adjustment for the number of CR exercise sessions attended and remained significant (p<0.001).”< span="">

 DISCUSSION

In addition to previously elaborated findings, authors also found that there were statistically significant differences in post-HTx VO2peak between patients with under 18 of CR sessions vs. those with >18 sessions. They quote the difference of 18.8±4.8 for >18 sessions vs. 16.4±4.3 for<18 sessions, p<0.01. While this is an undoubtedly better result, it would be worthwhile to discuss how these differences might truly impact on symptoms, patient well-being and quality of life and if there are works in the literature that comment on the magnitude of VO2peak changes and improvement in the quality of life. I think this would be a great addition to the discussion.

This important point has been further elaborated on in the Discussion section and the following has been added:

“Further, those who attended >18 sessions demonstrated a post-HTx VO2peak of 2.4 mL/kg/min higher than those who attended<18 sessions (i.e. 15% greater). Studies in HF have found that an increased VO2peak is associated with better outcomes (i.e. primary endpoints of all-cause hospitalizations and/or mortality) [6,7,35,36]. One such study found a 5% lower risk of all-cause mortality or hospitalization for every 6% increase in VO2peak, thereby highlighting the importance of even seemingly small improvements in VO2peak on long-term clinical outcomes [7].”

Study shortcomings and weaknesses

One of the more significant shortcomings of this work is the lack of post-transplant medication data used prior to the second measurement of VO2peak.  It is logical that medications used post-translation could indeed significantly impact on measured outcomes and this should be further emphasized as a limitation in the study. It should be clearly stated in Table 1. what do you mean by Medications. Is this prior to transplantation or drugs used after the transplantation prior to 2nd measurement? If medication intake refers to post-transplantation medication intake then regression analyses should be adjusted for medications. Please elaborate on this to greater detail.

We agree that medications may play a significant role in the outcomes of CPET analysis. The values shown in Table 1 are prior to HTx. This is stated in the legend of Table 1, however an additional symbol has been added to the Medications statement (*) to further direct the reader to the legend below. In addition, a statement in the discussion has been added to specify this shortcoming of the study:

“The distribution of post-HTx medications could potentially influence CPET measurements, therefore this should be taken into account as a limitation to the application of these results.

Another shortcoming is the missing data on donor information such as donor age and donor sex which could certainly impact clinical outcomes.

We also agree with this comment and the importance that donor information may have on these outcomes. As this data was unavailable for analysis in this study it is listed as a specific limitation yet encouraged for future follow-up studies as noted in the discussion section.

This is an important and well-conducted study on a fairly high number of cardiac transplant patients which is always a particularly vulnerable and somewhat exclusive population.

The authors would like to thank the reviewer for taking the time to review this manuscript and for the positive comment above.

I would advise authors to redo some of the statistical analyses and particularly assess the impact of a number of CR sessions (as a continuous or ordinal variable) on the VO2peak. It would be beneficial to know at what particular cut-off does CR yield maximum benefits. This cannot be ascertained if CR is incorporated into the regression model as a dichotomous variable. I would also advise inspecting the correlation between post-HTx VO2peak and number of CR sessions, similarly as reported for pre-HTx VO2 peak vs. post-HTx VO2 peak presented in Figure 1.

Thank you for this insight. The ROC curve was performed to account for this analysis and the unadjusted AUC was 0.692 with a p-value of<0.001. With sample size taken into account, the following CR exercise session values show as follows:

23 CR sessions: Specificity=0.804, Sensitivity=0.474, Sens-(1-Spec)=0.278

21 CR sessions: Specificity=0.735, Sensitivity=0.553, Sens-(1-Spec)=0.288

18 CR sessions: Specificity=0.549, Sensitivity=0.684, Sens-(1-Spec)=0.233

In line with the CR sessions analyzed in Figure 3 of the manuscript, this additional assessment has been added with the proper changes noted within the Methods and Results sections:

Methods – Statistical Analysis:

“The receiver operating characteristic (ROC) curve model was assessed to establish the area under the curve (AUC) for predicting quartile 4 (i.e. the highest VO2peak) and the cut-off for CR for maximal post-HTx VO2peak benefits.”

Results – Section 3.3

“The value of 18 sessions was also determined to be an appropriate cutoff for CR attendance necessary to maximize benefits in post-HTx VO2peak (AUC: 0.692, Specificity: 0.549, Sensitivity: 0.684, p<0.001). “

Reviewer 2 Report

Clinical and Rehabilitative Predictors of Peak Oxygen Uptake Following Cardiac Transplantation (HTX)

In a well-designed retrospective single centre study n=140 consecutive HTX patients have been investigated between 2007 and 2016 by cardiopulmonary exercise testing (CPET) prior and post HTX. Predictors of highest post HTX VO2peak were determined. The data revealed that pre-HTX VO2peak as well as attendance to structured cardiac rehabilitation prior and post HTX were significantly associated with an increased post HTX VO2peak. Cardiac rehabilitation followed a comprehensive approach and has sufficiently be described in the manuscript.

There are only minor comments:

-          From a total on n=204 HTX patients n=64 patients have been excluded. For transparency I would prefer to have a flow chart reflecting this selection process including in detail the exclusion criteria and the number of patients being affected

-          The paper would gain attraction, if the authors include a table summarizing the most important studies published so far and dealing with this topic (e.g. role of exercise pre- and post HTX). This table should include the major study criteria (study design, populations, interventions, controls, outcomes; PICOs). By this way we will gain a clear picture of the actual state of art and the role of the newly presented data.

-          Some comments should be given with respect to the evidence based minimal requirements of a successful cardiac rehabilitation of HTX patients. This is of importance for clinical practice, as probably all HTX patients should be transferred to cardiac rehabilitation, which not necessarily reflects actual clinical practice.

Author Response

Response to Reviewer 2 Comments

 We would like to thank reviewer #2 for their insightful and thorough review of our manuscript. We have carefully amended our manuscript based on the reviewer’s comments and appreciate the opportunity to improve the manuscript based on these insights. Please see our responses to the reviewer’s comments below:

 Clinical and Rehabilitative Predictors of Peak Oxygen Uptake Following Cardiac Transplantation (HTX)

In a well-designed retrospective single center study n=140 consecutive HTX patients have been investigated between 2007 and 2016 by cardiopulmonary exercise testing (CPET) prior and post HTX. Predictors of highest post HTX VO2peak were determined. The data revealed that pre-HTX VO2peak as well as attendance to structured cardiac rehabilitation prior and post HTX were significantly associated with an increased post HTX VO2peak. Cardiac rehabilitation followed a comprehensive approach and has sufficiently be described in the manuscript.

We appreciate your time in the review of this manuscript.

There are only minor comments:

-          From a total on n=204 HTX patients n=64 patients have been excluded. For transparency I would prefer to have a flow chart reflecting this selection process including in detail the exclusion criteria and the number of patients being affected

We appreciate this comment which was also brought up by Reviewer #1. We agree that additional clarification regarding the inclusion and exclusion of our patients is important. An additional description has now been added in the Methods section to provide a specific breakdown of the inclusion and exclusion of patients. Additionally, a flowchart has been added to the Methods section for additional clarity:

Figure 1. Flowchart for Patient Inclusion and Exclusion. Of the initially identified HTx patients, 54 patients lacked a pre-HTx or post-HTx CPET, 2 patients had incomplete CPET data, and 8 patients were lacking CR exercise session data, resulting in 140 patients for study analysis.

-          The paper would gain attraction, if the authors include a table summarizing the most important studies published so far and dealing with this topic (e.g. role of exercise pre- and post HTX). This table should include the major study criteria (study design, populations, interventions, controls, outcomes; PICOs). By this way we will gain a clear picture of the actual state of art and the role of the newly presented data.

This is an insightful comment. We agree with the important role that exercise plays specifically in the setting of HTx. The improvement in VO2peak following HTx in this study is in agreement with other exercise studies in the literature, yet the novelty of our results is specifically in the ability of CR participation to predict higher post-HTx VO2peak values. As such, our aim was to highlight other predictors of VO2peak that have been utilized in prior studies and demonstrate how this study is distinctly different (notably factors such as larger sample size and the new variable of CR). Because we agree with the reviewer’s perspective of placing our work in the context of previous literature, we now include a Table in the Discussion noting previous studies of particular importance in relation to the current manuscript:

“Table 5 provides information on four specific studies particularly relevant to the findings of the current study examining predictors of VO2peak in HTx patients.”

Table 5. Predictors of VO2peak   in Heart Transplant Patients

Study Group

N

Age (yrs)

Predictors of VO2peak

Time from Transplant

Post-Transplant VO2peak (mL/kg/min)

Douard et   al. 1997 [11]

85

52±12

Chronotropic reserve, time from transplantation, age of   donor, age of patient

1-100 months

21.1±6.0

Leung et   al. 2003 [13]

95

48±14

Age, sex, height, and weight (alternatively, body mass   index)

12 months

19.9±4.8

Nytrøen et   al. 2012 [18]

51

52±16

Muscular exercise capacity and body fat

1-8 years

Group 1: 23.1±3.7 Group 2: 32.6±4.4

Carvalho et   al. 2015 [14]

60

48±15

Age, sex, body mass index, heart rate reserve, and left   atrium diameter

64±54 months

unspecified

Note: VO2peak = Peak   oxygen uptake.   All data are presented   as Mean ± Standard Deviation unless otherwise specified.

-          Some comments should be given with respect to the evidence based minimal requirements of a successful cardiac rehabilitation of HTX patients. This is of importance for clinical practice, as probably all HTX patients should be transferred to cardiac rehabilitation, which not necessarily reflects actual clinical practice.

Thank you for this comment. The importance of CR in HTx has been further highlighted in the discussion, with specific emphasis on HTx-specific benefits as noted in the following addition:

“Specifically following HTx, the benefits of CR involvement are widespread for this clinically unique population; including counteracting marked deconditioning and skeletal muscle weakness associated with end-stage HF, corticosteroid treatment, and surgical recovery”

Reviewer 3 Report

I suggest minor revisions related to the following comments to the manuscript:

Line 14 - HTx - the abbreviation for cardiac transplant may have been explained in the abstract?

Line 18 - CPET standard protocol

Line 34 - Peak oxygen uptake (VO2peak) vs in note to Table 3  VO2peak = peak oxygen consumption. - same terms should be used.

Lines 40 and 45 - should it read post-HTx VO2peak instead of just VO2peak?

Line 52 - based off? Not based on? (sounds a bit odd to me)

Lines 48-53 - the hypothesis is based on references 15, 19 and 20

Line 59 - retrospective design? not retrospective cohort design?

Line 64 - unclear why only 140 of 204 patiens were included in the analysis when dealing with retrospective data? should the number 140 be number eligible (not included)?

Line 75 - purpose of validating research strategy - related to Line 76 - reviewed a random sampling of medical record Charts. Isn't this about monitoring the correctness of the data and not the Research strategy?

Line 77 - approved by the Mayo Clinic Institutional Review Board - instructions to authors in JMC ask for reference to the actual decicion number, refer to such number for this study?

Line s 76-78 - move the last sentence of the Clinical Characteristics section to the end of the Participants and Study Design section?

Lines 77-78 - all patients agreed to the use of their medical records for Research - did they sign written agreement?

Size of indents varies between paragrapfs or sections, looks a little messy. e.g. line 83 vs line 98.

Lines 84-85 - Medical records were examined to determine CR attendance specifically relating to postoperative HTx care versus CR for any cardiac-related event. What did you use this information for in the analysis? Are there any differences in CR between these indications?

Line 90 - what intensiy during the exercise sessions?

Line 100 - until volitional fatigue - relevance to setting/anxiety pre and post HTx?

Lines 123-124 - why were all these univariate variables chosen in the first place? Any justification or a fishing trip including everything from the patient record? factors involved in exercise capasity? explain those picked in model 3?

Line 133 - any corrections for multiple testing?

Line 135 - I guess Table 1 is pre-HTx values, but this is not written in the running text or table text. Point in time prior to HTx - should it be mentioned here? 

Table 2 - in note: VCO2 = Volume of carbon dioxide produced - unit is L/min  - a bit wrongly explained in the note?

Line 143 - extra empty line after subheading 3.2 - and Line 155 3.3 but not after 3.1 (line 135), looks messy.

Table 3 and 4 Peak Relative VO2peak (why this peak before relative?)  vs Table 2 Relative VO2peak (mL/kg/min) or just VO2peak (pre and post HTx) is it the same or any different? All variations of the term VO2 peak seems inconsistent, check and describe what is the same measure (unit) in the same tems, be precise.

Line 181 - Figure 1 - usualy figure text should be below figure (while only Table # should be above table) together With the figure text in Line 183 (shouldn't be split).

In Figure 1 - why not use the term relative before the units on the x and y-axes? (incosistency vs units in the tables?)

Line 188 and 190 should not be separated by the figure. Figure # and text should be below figure.

Figure 2 - is the mean and median the same? What about the number 18 itself? Not included? (<18 or="">18) 

Lines 168-172 - When CR exercise session attendance was divided by the median attendance of 18 sessions, those who attended >18 sessions had significantly higher post-HTx VO2peak compared to those with<18 sessions="" attended="">18 sessions:  18.8±4.8 vs.<18 sessions: 16.4±4.3, p<0.01) (Figure 2). - I don't understand the first half sentence before the comma? Or should it read ...was divided into two groups by the median attendance of 18...?

Table 4 heading should indicate that it is predictors of post-HTx VO2peak

Line 238 - 2.49 higher? times higher or what unit higher? VO2peak and mL/kg/min?

Author Response

Response to Reviewer 3 Comments

 We appreciate the time and effort Reviewer #3 invested in commenting on our manuscript. Below, we have responded to each of these comments and have subsequently made important changes to the manuscript based on these comments which we believe has led to a substantial improvement in the readability and clarity of our findings.

I suggest minor revisions related to the following comments to the manuscript:

We appreciate your time in the review of this manuscript.

Line 14 - HTx - the abbreviation for cardiac transplant may have been explained in the abstract?

Thank you, this has been adjusted to clarify this abbreviation.

Line 18 - CPET standard protocol

The sentence has been updated in the abstract.

Line 34 - Peak oxygen uptake (VO2peak) vs in note to Table 3 VO2peak = peak oxygen consumption. - same terms should be used.

Thank you. The footnote of Table 3 has been adjusted to state “uptake” to be consistent with the introduction.

Lines 40 and 45 - should it read post-HTx VO2peak instead of just VO2peak?

The authors have added “post-HTx” VO2peak to both lines 40 and 45 to further clarify this terminology.

Line 52 - based off? Not based on? (sounds a bit odd to me)

We agree and have updated this wording appropriately.

Lines 48-53 - the hypothesis is based on references 15, 19 and 20

These references have been added to the hypothesis sentence in the introduction:

“Based on previous studies on the relationship between CR involvement and VO2peak [15,19,20], we hypothesize that CR will surpass other predictive factors of post-HTx VO2peak in HTx patients.”

 Line 59 - retrospective design? not retrospective cohort design?

This wording has been updated to appropriately reflect the study design.

Line 64 - unclear why only 140 of 204 patients were included in the analysis when dealing with retrospective data? should the number 140 be number eligible (not included)?

We appreciate this comment and have clarified this in the Methods section.

Line 75 - purpose of validating research strategy - related to Line 76 - reviewed a random sampling of medical record Charts. Isn't this about monitoring the correctness of the data and not the Research strategy?

This wording has been adjusted accordingly to reflect the statement above.

Line 77 - approved by the Mayo Clinic Institutional Review Board - instructions to authors in JMC ask for reference to the actual decision number, refer to such number for this study?

IRB #15-007965 specifications have been added as follows:

“This study was approved by the Mayo Clinic Institutional Review Board (IRB #15-007965) and followed research authorization protocol for use of medical records as required by the state of Minnesota.”

Lines 76-78 - move the last sentence of the Clinical Characteristics section to the end of the Participants and Study Design section?

This sentence has been moved where requested.

Lines 77-78 - all patients agreed to the use of their medical records for Research - did they sign written agreement?

The authors have added “and followed research authorization protocol for use of medical records as required by the state of Minnesota” to the last sentence of the Participants and Study Design section.

Size of indents varies between paragraphs or sections, looks a little messy. e.g. line 83 vs line 98.

This inconsistency has been adjusted.

Lines 84-85 - Medical records were examined to determine CR attendance specifically relating to postoperative HTx care versus CR for any cardiac-related event. What did you use this information for in the analysis? Are there any differences in CR between these indications?

This is an important point to clarify. This was to ensure that the cardiac rehabilitation attendance for patients in this study was specifically related to their HTx procedure to accurately assess study objectives.

Line 90 - what intensity during the exercise sessions?

As heart rate is not considered an optimal method for determining exercise intensity following HTx due to denervation, the primary method for monitoring intensity following HTx was the rating of perceived exertion (RPE).

Line 100 - until volitional fatigue - relevance to setting/anxiety pre and post HTx?

We agree with the reviewers point that there may be additional factors involved in a patient’s exercise tolerance and the decision to terminate an exercise test, particularly in this unique population. An additional metric used to determine maximal effort during the exercise test included the respiratory exchange ratio which is provided in Table 2 (Pre-HTx: 1.15±0.13 and Post-HTx: 1.21±0.12).

Lines 123-124 - why were all these univariate variables chosen in the first place? Any justification or a fishing trip including everything from the patient record? factors involved in exercise capacity? explain those picked in model 3?

We appreciate this comment and the opportunity to clarify this point. The justification for each variable was based on several factors: 1) previous literature on predictors of VO2peak in various cardiac populations and/or specifically in the HTx population. 2) Known influencers of exercise capacity.

 Other factors were also obtained from the patient record initially, however we felt that the list provided in Table 3 most accurately reflects the potential variables involved in predicting post-HTx VO2peak and wanted to account for each in a univariate analysis:

-       Increasing age, female sex, and increasing BMI have all been shown to play a role in diminished peak exercise capacity.

-       Clinically history of disease (specifically diabetes, smoking, hypertension, and/or dyslipidemia) also may decrease peak values due to their influences on the respiratory system and/or overall vascular function.

-       Those patients in end-stage HF who receive LVAD therapy have been shown to have improvements in functional capacity throughout the 1st year following implantation.

-       The use of beta blocker medication has known chronotropic influence thereby influencing peak heart rate achieved.

-       The indication for HTx and varying forms of cardiomyopathy may impact peak exercise levels differently due to different pathologies.

-       Hemoglobin and Hematocrit provides an understanding of the oxygen-carrying capacity and delivery within the body.

-       White blood cell count provides a marker of inflammatory response and immune status, and negative correlations have been shown between WBC and VO2peak.

-       Creatinine provides a comprehensive assessment of renal function and HF status with higher values associated with lower VO2peak values.

-       Pre-HTx CPET data would be indicative of exercise capacity state prior to transplant

-       Cardiac Rehabilitation participation has been shown to improve VO2peak following HTx, as further described in the introduction and discussion sections.

Line 133 - any corrections for multiple testing?

There were no additional corrections for multiple CPET testing.

Line 135 - I guess Table 1 is pre-HTx values, but this is not written in the running text or table text. Point in time prior to HTx - should it be mentioned here? 

All data from Table 1 was obtained within 24 months prior to procedural date. This is stated in the Methods section and the additional statement has been added to the Results (3.1) to further clarify this information:

“All data shown in Table 1 was obtained from each patient within 24 months prior to procedural date.”

Table 2 - in note: VCO2 = Volume of carbon dioxide produced - unit is L/min - a bit wrongly explained in the note?

We have modified this note to “VCO2= production of carbon dioxide”

Line 143 - extra empty line after subheading 3.2 - and Line 155 3.3 but not after 3.1 (line 135), looks messy.

We have adjusted the formatting for consistency.

Table 3 and 4 Peak Relative VO2peak (why this peak before relative?)  vs Table 2 Relative VO2peak (mL/kg/min) or just VO2peak (pre and post HTx) is it the same or any different? All variations of the term VO2 peak seems inconsistent, check and describe what is the same measure (unit) in the same terms, be precise.

Thank you for noting this, these terms have been adjusted in Table 3 and 4 and “Peak” has been omitted. Additionally, the %Predicted variable in Table 2 has been edited to more accurately reflect the variable description.

Line 181 - Figure 1 - usually figure text should be below figure (while only Table # should be above table) together with the figure text in Line 183 (shouldn't be split).

These have been adjusted for all tables and figures.

In Figure 1 - why not use the term relative before the units on the x and y-axes? (inconsistency vs units in the tables?)

To optimize space and readability for font size, the axis titles have been left the same, however the word “Relative” has been added to the figure title to help clarify this information for the reader.

Line 188 and 190 should not be separated by the figure. Figure # and text should be below figure.

This has been updated.

Figure 2 - is the mean and median the same? What about the number 18 itself? Not included? (<18 or="">18) 

Thank you for this comment. Figure 2 represents the median of the CR exercise session data. The mean attendance was also 18 by rounding (17.53). The symbol within the text has been updated in the Results and Figure to reflect “≥” 18 CR exercise sessions.

Lines 168-172 - When CR exercise session attendance was divided by the median attendance of 18 sessions, those who attended >18 sessions had significantly higher post-HTx VO2peak compared to those with<18 sessions="" attended="">18 sessions:  18.8±4.8 vs.<18 sessions: 16.4±4.3, p<0.01) (Figure 2). - I don't understand the first half sentence before the comma? Or should it read ...was divided into two groups by the median attendance of 18...?

Thank you for pointing this out, the authors have added “into two groups” to the first half of this sentence for clarify.

Table 4 heading should indicate that it is predictors of post-HTx VO2peak

The addition of “Post-HTx” has been added to the title of Table 4.

Line 238 - 2.49 higher? times higher or what unit higher? VO2peak and mL/kg/min?

The units of “mL/kg/min” have been added to this sentence to explain the units of these study findings.

Round  2

Reviewer 1 Report

I would wish to congratulate authors for effectively attending all of my comments.
I advise the publication of this paper in its current form.